Unlocking insights: integrated text mining and interpretive structural modeling for enhanced user review analysis

Li Na 1486096069@qq.com 1
Liu Yu-Tao 2
Chen Zhan jslv123@126.com 3
1 School of Economics and Management, SouthWest Petroleum University , Chengdu , Sichuan , China
2 Higher Vocational and Technical Institute, Chengdu Neusoft University , Chengdu , Sichuan , China
3 School of Economics and Management, Southwest Jiaotong University , Chengdu , Sichuan , China
Alatas Bilal
Electronic publication date: 2024 Dec 23
Publication date: 2024
Volume: 10
Electronic Location ID: e2541
Received 2024 Jul 5; Accepted 2024 Nov 4
Copyright: ©2024 Li et al.
Copyright year: 2024
Copyright holder: Li et al.
License: This is an open access article distributed under the terms of the Creative Commons Attribution License, which permits unrestricted use, distribution, reproduction and adaptation in any medium and for any purpose provided that it is properly attributed. For attribution, the original author(s), title, publication source (PeerJ Computer Science) and either DOI or URL of the article must be cited.
License URL: https://creativecommons.org/licenses/by/4.0/

Keywords: User reviews, TF-IDF, Word2vec, Knowledge graph, ISM

Funding: The Research Centre for Fine Governance of Megacities, Key Research Base of Social Sciences in Sichuan Province No. TD2024Z06 The Sichuan Philosophy and Social Sciences Fund Project No. SCJJ23ND108 The Sichuan Mineral Resources Research Centre Project No. SCKCZY2023-YB001 The Resource-based City Development Research Centre No. ZYZX-YB-2411 This work was supported by the Research Centre for Fine Governance of Megacities, Key Research Base of Social Sciences in Sichuan Province (No. TD2024Z06), the Sichuan Philosophy and Social Sciences Fund Project (No. SCJJ23ND108), the Sichuan Mineral Resources Research Centre Project (No. SCKCZY2023-YB001), and the Resource-based City Development Research Centre (No. ZYZX-YB-2411). The funders had no role in study design, data collection and analysis, decision to publish, or preparation of the manuscript.

==============================
Effective keywords are extracted from the massive milk product user review data to construct thematic terms and explore the elemental influence relationships to assist manufacturers, and e-commerce platforms in understanding user behaviour and preferences and further optimise product design and marketing strategies. By fusing two different text mining methods, term frequency-inverse document frequency (TF-IDF) and Word2vec, we explore the semantic relationships, then visualise the relevance of user reviews by drawing knowledge graphs with Neo4j, and finally, be able to explore the relationship between the themes of the mined reviews, interpretative structural model (ISM) was used for a comprehensive evaluation, and the effectiveness of the method was verified on the Suning.com website dataset. The fusion of text mining and systematic analysis helps users to locate products quickly and precisely from the huge review information. The six elements of user reviews were categorized as freshness of taste, discounted prices, logistics, customer repurchase, product packaging, nutritional composition, and their element levels were divided into three layers. the first layer was discounted prices, customer repurchase, and logistics; the second layer was product packaging and nutritional composition; and the third layer was taste freshness.

Introduction

With the emergence of the Big Data era, the vast amount of resources and diverse forms of data on the Internet have posed a considerable challenge to information mining research. As a basic form of expressing users’ opinions and demands on products, the collection, mining, and analysis of user reviews have become an objective basis for e-commerce platforms to improve their service standards, enhance user satisfaction, as well as for manufacturers to solve user pain points and focus on user experience. How to quickly grasp the information of interest to users from the vast amount of user comment text, visualize user needs and dig deeper into the hidden connections in user comment information is not only an important technology that enterprises need to develop and improve but also a hot issue for scholars at present. Discovering the hot content users discuss based on text mining techniques and knowledge graphs is possible. At the same time, through the interpretative structural model (ISM), it is possible to find the interrelationships between the ranges, thus helping enterprises to explore user needs and determine the management direction.

In text mining, term frequency-inverse document frequency (TF-IDF) is suitable for simple text feature representation and word frequency statistics and is more effective for handling large-scale text collections (Toosi, Ghaaderi & Shokrani, 2021); Word2Vec is ideal for capturing the semantic and syntactic relationships of words, with better representation and inference capabilities and is suitable for language modelling and semantic-related tasks (Hong et al., 2022); ISM can describe the non-linear and non-symmetric relationships between multiple variables, and can better capture complex dependencies between variables, including non-linearities and interactions (Lin et al., 2022). Text mining methods can extract various information from the text to obtain a comprehensive description and understanding of the text (Kayikci, 2022). At the same time, explanatory structural modelling can further structure this information to reveal the relationships and dependencies between variables, further improving the accuracy and reliability of modelling and prediction and hence a better understanding of textual data. Thus, combining text mining and explanatory structural models can enhance overall performance, leading to more complete, accurate, and descriptive analysis results (Hattab, 2021).

In summary, unlike previous research methods that use a single method for text mining of user reviews, this study takes milk products on Suning.com website as the research object. It integrates word frequency analysis, relevance analysis, knowledge graph analysis and influencing factor analysis to mine key information in user reviews, i.e., word frequency analysis through TF-IDF to obtain keywords, relevance analysis through Word2vec to obtain word relevance, and visual display of the relationship of each word through Neo4j so as to classify the keywords into different categories, which in turn define the topics, and finally explore the relationship between topics through ISM. This study analyses the content of user discussions, areas of interest and interaction with products to better meet user needs and improve the quality of e-commerce platform services and manufacturers.

Literature Review

The mining of user review data has attracted many scholars because of its excellent research value and relevance. Much progress has been made in the research on comment data mining. The two main aspects of mining comment data are word frequency and systematic analysis.

In word frequency analysis, developing keyword extraction methods is a continuous process of innovation. Early in the field, Justeson & Katz (1995) proposed the TERMS system, which is based mainly on the distinguishing features of terms from ordinary words one by one technical terms are almost always composed of nouns and adjectives, as well as a small number of prepositions connecting two noun phrases. Liu, Teller & Friedman (2004) introduced the concept of association rules in natural language processing. Firstly, user comments were divided into words and lexical annotations, and then noun verbal terms were extracted using the technique of frequent item mining. Then, adjectives associated with the nouns were removed according to association rules, summarizing the item’s features. The aim was to understand user reviews better and thus extract critical parts of the things (Hu & Liu, 2004). Scaffidi et al. (2007) designed and implemented the Red Opal search system. The system removes product features using TF-IDF and lexical annotation methods and performs sentiment analysis on each segment. Li, Cui & Ji (2015) proposed a keyword extraction algorithm based on the Word2vec model. They used Word2vec for word separation, word embedding, and clustering by calculating the similarity between words to obtain text keywords. Experiments demonstrated that the algorithm’s keyword extraction accuracy was significantly better than others in longer texts. Cerisara, Krl & Lenc (2018) applied Word2vec to conversational behavior recognition and demonstrated that deep neural networks outperformed maximum entropy classifiers. Experimental results showed that the model could better capture the semantic features of spoken commentary text in the keyword extraction phase and achieved significant improvements in other baselines.

For systematic analysis, the similarity is calculated based on the corpus, i.e., based on the information in a more extensive set of texts, and this type of approach evaluates ’semantic’ similarity. Semantic similarity is obtained through statistical analysis of many documents, where the “large number of documents” is the corpus. Marom & Zukerman (2009) explored corpus-based approaches to automating email responses, focusing on information-gathering techniques and granularity. We evaluated different methods and found that they were successful in automatically generating most of the email responses. The meta-selection process was also investigated to provide a unified response automation solution for new query emails. Hashemi & Shakery (2014) proposed a method for mining high-quality translation knowledge from UTPECC, using term association networks (TAN) to extract translation knowledge. A post-processing step is used to detect mistranslated terms. The evaluation results show that the use of the UTPECC approach is significantly better than the dictionary-based approach and is particularly suitable for translating extra-lexical terms and extended query words; Tutubalina et al. (2017) conducted a study for different therapeutic groups of drugs to find compounds with potentially similar biological effects. By applying the Word2vec model, it was possible to model and analyze the semantic relationships between drugs, further revealing the similarity of their possible physical results; Qiu et al. (2019) combined a plain Bayesian model with Word2vec and experimentally demonstrated that this scheme was more efficient than TF-IDF, Text Rank and Rapid Automatic Keyword Extraction (RAKE), with a substantial improvement in F1 values.

In summary, the above provides a history of developing some keyword extraction methods, from early plans based on word frequency and location information to later ways that introduce techniques such as association rules and deep learning. However, text mining for information extraction of user comments still needs to improve, such as ignoring textual and contextual information, lack of accuracy in keyword selection, and poor extraction accuracy due to using a single method, which affects the final information extraction results. Therefore, this paper will be oriented to solving practical problems by considering the textual features and semantic similarity of the comment data, obtaining keywords of user comment texts through TF-IDF, conducting relevance analysis on the accepted keywords, displaying all word relationships, and constructing themes by drawing knowledge graphs through Neo4j, and finally treating each theme as an element and analyzing the influence and hierarchical relationship between the elements The knowledge graph is then used to analyze the impact and hierarchical relationships between the details, so that the information in the user reviews can be more comprehensively mined to understand better the user’s evaluation and demand for the product or service, providing a valuable reference for e-commerce platforms and manufacturers to make decisions and improvements.

Research Methodology and Empirical Analysis

Portions of this text were previously published as part of a preprint (https://www.researchsquare.com/article/rs-3375941/v1). The website was launched in December 2010, covering home appliances, department stores, mother and child, and books. In this paper, the user reviews of milk products on the Suning Tesco website are crawled, the crawled user reviews are stored in CSV format, and the user reviews are Chinese sub-phrased and used for subsequent data analysis. The overall process is shown in Fig. 1. This paper’s research idea will follow the logic of “word frequency analysis → correlation analysis → correlation visualisation → elemental relationship analysis” for empirical analysis.

Figure 1 Research ideas.

This paper’s research idea will follow the logic of “word frequency analysis → correlation analysis → correlation visualisation → elemental relationship analysis” for empirical analysis.

Research Methodology

(1) Word frequency analysis. The former is the ratio of the number of occurrences of a specific word in the user reviews of milk products to the total number of words in the text of user reviews, and the latter is the ratio of the number of user reviews of all milk products to the number of studies containing a specific word, and finally, the words with the highest word frequency will be used as keywords in user reviews;

(2) Relevance analysis. Since TF-IDF cannot reflect contextual information, the keywords calculated by TF-IDF need to be analyzed for relevance, i.e., Word2vec is used to measure the similarity between words to obtain semantic relationships. Firstly, the expression vector of each keyword is calculated, according to which the spatial distribution of each word can be obtained, and then the related terms of each keyword can be found;

(3) Visualisation of the correlation situation. The correlation analysis can obtain the correlation between individual words but cannot know the spatial relationship of unique words in the overall review. Based on the logical connection between words and relevance, comments can be considered nodes and relevance as relationships. Based on this idea, the relevance of keywords can be visualized through the Neo4j graphical database, which can clearly and intuitively show the classification of the content of user comments and define different themes for each category containing other words according to the variety;

(4) Element relationship analysis. The different themes obtained through the data visualization are considered content elements reflecting the user’s needs, and the interrelationships and hierarchies between the components are explored using the ISM systematic analysis method.

Data acquisition

This experiment is based on Python 3.7, and the data is based on user reviews of milk products on Suning.com official website of Suning.com, and the target data is obtained through a self-written web crawler. Since the target website is a dynamically loaded web page, capturing the page information through the requests module is impossible, so it is necessary to use Selenium to automatically simulate the operation to crawl the HTML tags of the web page (Chen et al., 2021).

In this study, we crawled 60,367 comments from Yili, Mengniu, Junlebao, Gandhi Farm, Huang’s Dairy, Shengmu, Anjia, and Deya milk products on the official website of Suning.com, excluding invalid comments such as default comments, blank comments, and garbled text, and finally obtained 10,273 valid user comments.

Before formal text analysis, the text of the comments needed to be word-separated using jieba word-separation. To achieve more accurate word separation results and subsequent text analysis accuracy, punctuation and useless characters (numbers, tone words, etc.) were deactivated, i.e., these results were word filtered so as not to affect keyword extraction and word relevance analysis. The data for the first 50 user comments are shown in Table 1. The top 50 user comments on the milk product on the Suning.com website, as crawled by the crawler, reveal users’ different attitudes towards the product’s consumption.

Table 1 User comments form.

The top 50 user comments on the milk product on the Suning.com website, as crawled by the crawler, reveal users’ different attitudes towards the product’s consumption.

Serial number	Splitting results	Serial number	Splitting results	
1	The packaging design is atmospheric and the milk flavour is rich and good to taste. One praise!	26	I’ve been drinking this brand of milk, children love it, especially convenient	
2	Very fast delivery, children love to drink	27	The pro futures at the end of July, the taste is not bad.	
3	Special satisfaction, Suning DS	28	Still good, have bought twice	
4	This milk purchase is more cost-effective, before the child also said the taste is good, so decisive purchase	29	The milk is very rich and the date is fresh	
5	The milk has been repurchased, drink at ease very good.	30	It’s very good, very fast, arrived home fresh, good stuff ah!	
6	The old brand. I’ve been drinking it for a long time.	31	Good price, fresh date, very fast logistics	
7	It’s good value for money and worth buying!	32	As always, I always drink and buy Amush.	
8	The milk is good, it’s rich and creamy!	33	Good good good good good good good good good good good	
9	The original milk with no additives	34	The taste is okay, the price is affordable!	
10	I bought it for the New Year’s Day, it’s economical and good for people	35	This is still very good, the production date is April 2022. I still feel satisfied with it, it is the original yogurt. The express delivery is also okay, the speed is also still relatively fast, still relatively satisfied, or indeed can, very recommendable to buy, really good it. Still more satisfied	
11	The yogurt tastes very good, ah, the date is also quite fresh, S.F. super power, delivered to the yogurt is also still very fresh, family members are very like, praise praise!	36	quite thick, there are activities with coupons to use when more affordable	
12	The banana-flavored yogurt from Gandhi Ranch is mellow and delicate, and the whole family loves it!	37	Good milk, will re-purchase many times, praise.	
13	Very good, worth buying, next time continue	38	The pure milk is very nutritious and good to drink, children every day a bottle	
14	The kids love to drink it	39	The milk was good, heated up and had a milk skin.	
15	It was very fast. It is the same as the supermarket.	40	The price is also affordable, praise a!	
16	Sweet milk, tasty and freshly dated	41	The old family milk, very like to drink	
17	Cheap and very good ! ! ! !	42	The date is very fresh, the important thing is so cheap	
18	Creamy, smooth and thick, repurchased several times	43	Help parents bought directly sent home, parents are very happy old people also need to supplement calcium, we remember to care for children, but also need to care for parents! I didn’t take a picture of the real thing, I took a picture of it, but I still give credit to Suning.com, and the delivery was fast!	
19	Very good! And it’s all sent out S.F. now?	44	Very good	
20	Rich flavour, fresh date and easy to shop.	45	Drinkable	
21	The goods are good and worth buying!	46	Junlebao yogurt, flavorful, fresh date, Suning Yi purchase price is more affordable, delivery on better.	
22	The actual fact is that you can find a lot of people who have been in the marketplace for a long time, and they’ve been in the market for a long time.	47	Delicious and creamy!	
23	Good Oh activity is really powerful	48	Fresh, flavorful, fresh date, very good service.	
24	The yogurt quality is very good, the date is new, good to drink, the whole family like to drink, praise	49	Buy ........................ buy	
25	Our family has always been drinking the Amosi good to drink	50	Very awesome, super fast logistics, genuine!!!	

Word frequency analysis

The jieba.analyze.extract_tags () function was used on the splitting results to obtain the keywords in the comment text, with the top K parameter set to 20, indicating that the top 20 keywords with the most significant TF-IDF weight values were returned (Budel et al., 2023), and the results are shown in Table 2. As seen from Table 2, users’ concerns about milk products can be roughly divided into elements, including taste, freshness, logistics and price, which are also user needs for food products. Based on the TF-IDF values in Table 2, the keywords in the results of this word frequency analysis were output as a word cloud diagram (Fig. 2).

Table 2 User comments keywords.

The jieba.analyze.extract_tags () function was used on the splitting results to obtain the keywords in the comment text, with the top K parameter set to 20, indicating that the top 20 keywords with the most significant TF-IDF weight values were returned.

Ranking	Keywords	TF-IDF	Ranking	Keywords	TF-IDF	
1	Delicious	0.3568	11	Price	0.0775	
2	Date	0.2476	12	Full-Bodied	0.0766	
3	Flavour	0.1834	13	Quick	0.0745	
4	Fresh	0.1791	14	Express Delivery	0.0724	
5	Mouthfeel	0.1484	15	Pure Milk	0.0663	
6	Affordable	0.1231	16	Repurchase	0.0662	
7	Logistics	0.1199	17	Many Times	0.0621	
8	Packaging	0.1153	18	Taste	0.0495	
9	Shipping	0.1006	19	Cheap	0.0487	
10	Yoghurt	0.0906	20	Value For Money	0.0471	

Figure 2 Milk products keyword word cloud.

Clearly shows the frequency of keywords, especially those that appear most frequently in the comments.

The word cloud diagram in Fig. 2 clearly shows the frequency of keywords, especially those that appear most frequently in the comments. For example, “delicious” and “taste” indicate that the main focus of customers on milk products is on the taste of the product itself, while “date” and “fresh” point to the freshness of the product, while “price”, “cheap” and “affordable” point to the price and benefits of the product itself. These are the core content that users are most concerned about, so e-commerce platforms or manufacturers can use the word cloud to dig deeper into users’ demands and ideas and also to give users a general understanding of their products so that they can quickly refine product information before making their own purchasing decisions.

Relevance analysis

To obtain the interrelationships between the keywords, i.e., to get the relevant words for each keyword, Word2vec was used for the relevance analysis. In this section, the Word2vec model is invoked for training, with default values used for most of the model parameters and other parameters set (Table 3).

As shown in Table 4, the word vector of each word shows its position in space, and the phrases most closely related to each keyword can be analyzed according to their distance, i.e., the terms related to each keyword can be calculated from the word vector (Heimerl & Gleicher, 2018).

Table 5 indicates the degree of relationship with the keyword-related words, and the higher the value, the closer the relationship with the word, i.e., closer to the meaning or pointing content of the keyword (Benito-Santos & Sanchez, 2019; Radhakrishnan et al., 2017). The related words for each keyword are shown in it. For example, the associated terms for the keyword good to drink are very fragrant, fragrant, fabulous, indeed, sour, and sweet, where the most relevant observation is very aromatic, reaching 0.7561, indicating that users think milk is good to drink mainly from the fragrance of milk (Fig. 3). To show the relationship situation of all words more visually, the relevance situation of all keywords was visualized by drawing a knowledge graph through Neo4j.

Table 3 Word2vec parameter setting.

To obtain the interrelationships between the keywords, i.e., to obtain the relevant words for each keyword, Word2vec was used for the relevance analysis. In this section, the Word2vec model is invoked for training, with default values used for most of the model parameters and other parameters set.

Parameters	Meaning	Value	Value basis	
vector_size	Word vector dimension	20		
window	Maximum distance allowed between words and predicted words	2	The distribution hypothesis proposed by Harris (1954) states that [5], taking values of [1,5] achieves a uniform distribution	
min_count	Minimum threshold for words	3	[0,100] is more reasonable	
negative	Sample size for negative sampling	10	Small-scale data taking [5,20]	
topn	Output the number of words with the highest relevance to the target word	5		

Table 4 User reviews keyword word vector.

The word vector of each word shows its position in space, and the phrases most closely related to each keyword can be analyzed according to their distance, i.e., the terms related to each keyword can be calculated from the word vector.

Keywords	Word vectors	
Delicious	[−0.6920	−0.2668	0.1874	−0.8941	−0.5032	−1.2193	
0.7288	−1.4380	0.8745	−0.7369	−1.0266	−1.6656	
−0.3567	−0.8803	1.9274	−1.3176	1.0117	−0.5044	
0.8275	−0.0159]					
Date	[−1.1384	−1.5294	1.8028	2.1135	−0.6066	−0.8862	
−0.3928	0.1051	−1.4157	−0.8966	−0.3170	−1.6074	
−1.9676	−1.2788	2.3133	−1.2268	0.2742	1.2267	
−1.2418	−0.6370]					
Flavour	(1.6519)	−1.5223	0.7011	−0.2965	−0.0714	−0.4816	
0.2895	−1.1843	0.2961	−0.6455	0.7143	−3.2508	
0.0128	−1.8315	1.5518	−1.8104	0.3513	0.7882	
0.3918	−0.1321]					
Fresh	[−1.6040	−2.3627	2.3166	0.4502	−0.1624	−0.2030	
1.1098	1.0415	−0.9931	−0.3903	−1.6381	−1.6145	
−1.4940	−0.9870	1.6570	−0.6893	0.8794	−0.8505	
−0.1580	−0.0888]					
Mouthfeel	[−1.9374	−2.6161	1.1871	0.1442	−0.2308	0.2078	
0.3960	−0.9050	−0.6053	−0.9586	0.8296	−2.0282	
0.2411	−1.7597	1.1260	−0.9439	1.1340	0.9714	
0.5033	−0.6855]					
Affordable	(0.9750)	−0.6701	0.9544	0.6316	−1.0882	−1.9741	
2.9998	1.6253	−0.1947	−2.2018	0.7940	−3.0239	
−0.3342	−0.1411	2.2813	0.3486	0.7932	−0.5954	
1.2828	−0.8490]					
Logistics	[−1.2479	0.4860	0.6396	3.9474	−1.5368	0.4153	
0.0849	1.9826	−1.8231	−1.0963	1.4801	−2.1907	
−0.3132	−3.1924	0.7379	0.9645	−2.1234	0.2577	
0.3876	−1.0834]					
Packaging	−2.9240	−3.4385	0.1487	0.5023	−0.5255	−1.1438	
[-0.4577	0.0294	−3.1107	0.4715	1.3253	−0.7490	
−1.1457	−4.2402	−0.9431	0.0352	−0.3170	−0.8106	
−0.7209	−0.4160]					
Shipping	(1.1172)	−0.0130	0.6298	3.1526	−3.0921	−1.1576	
0.8945	1.7930	−2.4505	−0.8090	−0.3847	−2.2052	
−2.3654	−4.3214	1.3561	1.8756	−0.7713	1.4122	
−1.1096	−1.0064]					
Yoghurt	[−1.9703	−1.4077	1.0407	−0.2978	−0.7383	0.2700	
−1.7449	−2.9350	0.3295	0.2144	−1.7305	−1.9597	
−1.8824	−0.2530	0.0531	−0.6260	0.3436	−0.6049	
1.4202	−1.0494]					
Price	[−2.1319	−0.4495	−0.2058	0.9452	0.1980	0.1336	
0.1300	1.0743	−2.5834	−2.9149	−1.0999	−2.4789	
0.0068	−0.2607	1.5269	−0.0269	−0.6963	0.9388	
2.7880	−2.0117]					
Full-bodied	[−4.3586	−2.7024	2.3281	3.1169	−1.7908	−0.2993	
0.3643	−0.7814	−3.1217	1.1516	−1.3400	0.2899	
1.1456	−0.8703	2.6749	−1.5490	1.3605	0.1313	
1.0740	1.9265]					
Quick	(0.4304)	−1.5430	1.3921	−0.9669	1.0883	−3.1407	
−1.6129	0.1777	−2.1707	−0.2330	−0.2973	−1.0211	
−2.7294	−2.9387	1.3056	2.1994	0.6401	−1.2058	
−0.6120	−0.8508]					
Express delivery	[−3.0181	1.2122	−0.4015	5.5648	−2.2107	−0.5137	
−1.6255	3.3593	−1.6447	0.6221	−0.5689	0.7878	
−0.2562	−3.0700	2.3077	0.3203	−1.9505	−2.3840	
−1.5368	−0.6940]					
Pure milk	[−1.7936	−2.4432	−0.5423	−2.4205	2.1436	0.7165	
−2.9969	−2.3708	−0.0059	−1.7267	0.1802	−0.9618	
0.1011	1.4304	2.3570	−1.1008	2.5010	−2.0630	
2.8196	2.2313]					
Repurchase	[−5.1592	2.6877	3.3645	−1.7283	1.0239	−0.4188	
1.8482	0.4781	2.5435	−1.4603	−0.4948	1.2099	
0.4250	−2.6295	1.5306	−1.5252	−0.0393	1.9480	
1.4819	0.3295]					
Many times	(1.0438)	0.8622	3.2431	−0.1134	−1.0965	−0.1818	
2.2224	−1.7466	−1.8196	−1.3382	−0.4418	1.0194	
−3.5827	−0.1511	1.7202	−1.5122	3.0671	2.9381	
0.2620	1.1779]					
Taste	[−2.1463	−2.8672	1.0249	−1.0585	1.2043	−0.5950	
1.0592	−3.7165	1.0586	−0.6063	0.0896	−1.8911	
−1.4456	−2.4598	0.2484	−1.3719	1.9736	−0.3180	
−1.0923	−1.2686]					
Cheap	(0.6931)	0.7872	−1.2885	−0.0366	−1.6105	−2.2567	
3.6182	2.2658	1.5676	−1.8635	0.3790	−2.8954	
−0.4266	0.9024	3.3743	2.6467	2.0060	0.6304	
0.4459	−2.2237]					
Value for money	[−0.2654	−1.6181	3.1804	1.0337	−0.1359	1.7024	
5.5274	1.4372	−3.0442	−4.8828	0.9330	−1.8727	
0.9065	−2.2762	−1.5181	−3.6319	0.6312	−0.1488	
−0.1082	0.7111]					

Table 5 Keyword distribution in user comments section.

The table indicates the degree of relationship with the keyword-related words, and the higher the value, the closer the relationship with the word, i.e., closer to the meaning or pointing content of the keyword.

Keywords	Related words	Relevance	Keywords	Related words	Relevance	
	Very Fragrant	0.7561		Economical	0.8000	
	Fragrance	0.7093		Prices	0.7076	
Delicious	Wonderful	0.6720	Price	Awesome	0.6521	
	Indeed	0.6691		Events	0.6501	
	Sour and sweet	0.6473		Product prices	0.6026	
	Date of production	0.8633		Mellow and thick	0.8239	
	Freshness	0.6997		Strong aroma	0.8040	
Date	Taste	0.6962	Full-Bodied	Pure	0.7750	
	Milk quality	0.6882		Finesse	0.7612	
	Shelf life	0.6773		Sweet and savory	0.7452	
		0.9093		Quick	0.8709	
	Mouthfeel	0.8988		Ultrafast	0.8208	
Flavour	Milk quality	0.7618	Quick	In time	0.7352	
	Creaminess	0.7386		Rapid	0.7316	
	Delicious	0.6699		More Satisfaction	0.7223	
	Not bad	0.7005		Distribution	0.8405	
	Satisfaction	0.6858		Transport	0.7940	
Fresh	Date	0.6675	Express delivery	Delivery to your door	0.7922	
	Strong aroma	0.6642		Logistics	0.7547	
	Pure	0.6584		Shipping	0.7282	
	Flavour	0.8538		High in calcium	0.7809	
	Milk quality	0.7959		Softened Milk	0.7788	
Mouthfeel	Delicious	0.7521	Pure milk	Yoghurt	0.7056	
	Taste	0.7480		Sour milk	0.6816	
	Very fragrant	0.6982		Suitable for people	0.6609	
	Offers	0.8695		Repeat purchases	0.8515	
Affordable	Suitable	0.7730	Repurchase	Patronage	0.7720	
	Cheap	0.7636		Purchase	0.7617	
	Value for money	0.6993		Bought	0.6332	
	Bargain buy	0.6793		Later	0.5892	
	Distribution	0.8723		Second	0.9120	
	Shipping	0.8653		Many a time	0.7044	
	Speeds	0.7834		Numerous times	0.6978	
Logistics	Delivery to your door	0.7787	Many times	Long-term	0.6848	
	Courier	0.7557		Regular Clients	0.6869	
	Milk crate	0.8296		Flavour	0.8709	
	Careful	0.7115		Mouthfeel	0.7276	
Packaging	Strictly	0.7115	Taste	Delicious	0.6600	
	Packaging	0.6766		Products	0.6263	
	Serving size	0.6696		Very fragrant	0.6276	
	Transport	0.8612		Offers	0.7974	
	Logistics	0.8594		Value for money	0.7478	
Shipping	Distribution	0.8316	Cheap	Affordable	0.7373	
	Delivery	0.7999		Coupons	0.6617	
	Delivery to your door	0.7449		Events	0.6479	
	Milk	0.7573		Nutritional value	0.7656	
	Pure milk	0.7120		Calcium milk	0.6794	
Yoghurt	Sour milk	0.6958	Value for money	Nutritional composition	0.6688	
	Authentic	0.6879		Contains calcium	0.6517	
	Products	0.6700		Content	0.6404	

According to Table 6, classification of the results of the relevance analysis of user comments. Based on the results of the keyword relevance analysis in Fig. 3, it can be seen that the user comments on the items are divided into eight sections.

Analysis of influencing factors

By analyzing Table 6, it can be seen that Category 3 and Category 5 both described content in terms of logistics and express delivery and, therefore, could be classified into one category; while Category 4 and Category 7 both referred to the situation where consumers trusted the shop and thus returned to it repeatedly, and therefore could also be classified into one category. The above analysis was combined, and Table 6 was rearranged to finally organize the user review content into six categories and group the categories into different themes based on the range of the keywords included (Gonalves et al., 2023), with the complete results shown in Table 7. The six subject terms collated as the six elements of user comments are analysed by the ISM explanatory structural model to investigate the influence between the elements.

As shown in Table 8, drawing on the findings of Stan et al. (2010) and others, the following can be specified: freshness of taste and discounted prices offer both influence customer repurchase, while customer repurchase, logistics and product packaging all influence discounted prices of the product. In addition, discounted price and product packaging affect logistics, freshness of taste directly affects nutritional composition. this shows the adjacency matrix of the model, the initial input matrix, abstracted from the relationships between the elements.

Figure 3 Milk product keyword relevancy analysis.

To show the relationship situation of all words more visually, the relevance situation of all keywords was visualized by drawing a knowledge graph through Neo4j.

Table 6 Classification of results of correlation analysis of user comments.

It was found that Category 3 and Category 5 both described content in terms of logistics and express delivery and, therefore, could be classified into one category. Category 4 and Category 7 both referred to the situation where consumers trusted the shop and thus returned to it repeatedly, and therefore could also be classified into one category.

Category	Contains words	
1	Pure Milk, Softened Milk, Suitable for People, High Calcium, Sour Milk, Pure Milk, Authentic, Yogurt, Product, Delicious, Great, Very Fragrant, Taste, Sweet and Sour, Fragrant, Indeed, Taste, Milk Carton, Milk Quality, Delicious, Date, Shelf Life, Production Date, Satisfied, Fresh, Good, Strong, Pure, Mellow, Delicate, Rich, Sweet And Delicious	
2	Affordable, Cheap, Good Value, Suitable, Discount, Bargain, Item Price, Event, Price, Coupon, Economy, Awesome, Price	
3	Logistics, Express, Speed, Home Delivery, Delivery, Distribution, Delivery, Dispatch	
4	Many Times, Long Term, Regular Customer, Many Times, Second Time, Countless Times	
5	Very Fast, Quick, Prompt, Superfast, More Than Satisfactory, Prompt	
6	Packaging, Tight, Careful, Portions, Boxes, Milk Cartons	
7	Repurchase, Buy, Bought, Repurchase, Future, Patronage	
8	Value For Money, Nutritional Composition, Content, Calcium Milk, Nutritional Value, Calcium Content	

Table 7 User comments categories.

The six subject terms collated as the six elements of user comments are analysed by the ISM explanatory structural model to investigate the influence between the elements.

Category	Contains words	Topics	
1	Pure Milk, Softened Milk, Suitable for People, High Calcium, Sour Milk, Pure Milk, Authentic, Yogurt, Product, Delicious, Great, Very Fragrant, Taste, Sweet and Sour, Fragrant, Indeed, Taste, Milk Carton, Milk Quality, Delicious, Date, Shelf Life, Production Date, Satisfied, Fresh, Good, Strong, Pure, Mellow, Delicate, Rich, Sweet and Delicious	Freshness of taste	
2	Affordable, Cheap, Good Value, Suitable, Discount, Bargain, Item Price, Event, Price, Coupon, Economy, Awesome, Price	Discounted prices	
3	Logistics, Express, Speed, Home Delivery, Delivery, Distribution, Delivery, Shipping, Very Fast, Quick, Timely, Superfast, More Than Satisfactory, Rapid	Logistics	
4	Many Times, Long Term, Regular Customer, Many Times, Second Time, Countless Times, Repurchase, Buy, Bought, Repurchase, Future, Patronage	Customer Repurchase	
5	Packaging, Tight, Careful, Portions, Boxes, Milk Cartons	Product Packaging	
6	Value For Money, Nutritional Composition, Content, Calcium Milk, Nutritional Value, Calcium Content	Nutritional composition	

Table 8 Constructing the adjacency matrix.

Drawing on Stan et al. (2010) and others, the following can be specified: freshness of taste and discounted prices offer both influence customer repurchase, while customer repurchase, logistics and product packaging all influence discounted prices of the product. In addition, discounted price and product packaging affect logistics, freshness of taste directly affects nutritional composition. This shows the adjacency matrix of the model, the initial input matrix, abstracted from the relationships between the elements.

Key elements	Freshness of taste	Discounted prices	Logistics	Customer repurchase	Product packaging	Nutritional composition	
Freshness of taste	0	1	0	1	0	1	
Discounted prices	0	0	1	1	0	0	
Logistics	0	0	0	1	0	0	
Customer repurchase	0	1	0	0	0	0	
Product packaging	0	1	1	0	0	0	
Nutritional composition	0	0	0	1	0	0	

According to Table 9, the adjacency matrix multiplication of the model, which is an intermediate computational procedure to obtain the reachable matrix by successively multiplying the adjacency multiplication matrix until the matrix does not change (Mallick, Chaudhari & Joshi, 2022).

Table 9 Adjacency matrix multiplication.

The adjacency matrix multiplication of the model, which is an intermediate computational procedure to obtain the reachable matrix by successively multiplying the adjacency multiplication matrix until the matrix does not change.

Key elements	Freshness of taste	Discounted prices	Logistics	Customer repurchase	Product packaging	Nutritional composition	
Freshness of taste	0	1	0	1	0	1	
Discounted prices	0	0	1	1	0	0	
Logistics	0	0	0	1	0	0	
Customer repurchase	0	1	0	0	0	0	
Product packaging	0	1	1	0	0	0	
Nutritional composition	0	0	0	1	0	0	

As shown in Table 10, with the adjacency matrix representing the direct relationship between elements and the accessibility matrix representing whether the transmission between elements leads to an indirect influence relationship.

Table 10 Reachability matrix.

With the adjacency matrix representing the direct relationship between elements and the accessibility matrix representing whether the transmission between elements leads to an indirect influence relationship.

Key elements	Freshness of taste	Discounted prices	Logistics	Customer repurchase	Product packaging	Nutritional composition	
Freshness of taste	0	1	0	1	0	1	
Discounted prices	0	0	1	1	0	0	
Logistics	0	0	0	1	0	0	
Customer repurchase	0	1	0	0	0	0	
Product packaging	0	1	1	0	0	0	
Nutritional composition	0	0	0	1	0	0	

Table 11 shows the hierarchical decomposition process of the model. At each extraction, if the intersection of the reachable and reachable sets with the initial stage is the same, the element set can be extracted for hierarchical decomposition (Fernandes, Simsek & Kantarci, 2021). Table 12 shows the hierarchy of models, where the top level is the system’s ultimate goal, while the following levels are the reasons for the upper levels respectively.

Table 11 Extraction of results process.

The table shows the hierarchical decomposition process of the model. At each extraction, if the intersection of the reachable and reachable sets with the initial stage is the same, the element set can be extracted for hierarchical decomposition.

Results after 1st draw	
Key elements	Accessible collection	Reachable sets and prior sets	
Freshness of taste	Freshness of Taste, Discounted Prices, Logistics Customer Repurchase, Nutritional Composition	Freshness of Taste	
Discounted prices	Discounted Prices, Logistics, Customer Repurchase	Discounted Prices, Logistics, Customer Repurchase	
Logistics	Discounted Prices, Logistics, Customer Repurchase	Discounted Prices, Logistics, Customer Repurchase	
Customer repurchase	Discounted Prices, Logistics, Customer Repurchase	Discounted Prices, Logistics, Customer Repurchase	
Product packaging	Discounted Prices, Logistics, Customer Repurchase, Product Packaging	Product packaging	
Nutritional composition	Discounted Prices, Logistics, Customer Repurchase, Nutritional Composition	Nutritional Composition	
Results after the 2nd draw	
Key elements	Accessible collection	Reachable sets and prior sets	
Freshness of taste	Freshness of Taste	Freshness of Taste	
Product packaging	Product Packaging	Product Packaging	
Nutritional composition	Nutritional Composition	Nutritional Composition	
Results after 3rd draw	
Key elements	Accessible collection	Reachable sets and prior sets	
Freshness of taste	Freshness of taste	Freshness of taste	

Table 12 Element hierarchy.

The table shows hierarchy of models, where the top level is the system’s ultimate goal, while the following levels are the reasons for the upper levels, respectively.

Levels	Key elements	
Level 1	Discounted Prices, Logistics, Customer Repurchase	
Level 2	Product Packaging, Nutritional Composition	
Level 3	Freshness of Taste	

A directed connectivity diagram based on the results of the hierarchy can show the scale of the model more visually (Fig. 4).

Figure 4 Hierarchy diagram.

A directed connectivity diagram based on the results of the hierarchy can show the scale of the model more visually.

User review results analysis and route optimisation

Analysis of results

By analyzing the user reviews of milk products on Suning.com, the data were mined, analyzed, and comprehensively evaluated through three methods, TF-IDF, Word2vec, and ISM, in turn. The above three methods could drill a large amount of characteristic information, interrelationships, and the influence between each element from the user reviews, thus e-commerce platforms, producers, and other system management inspiration and decision-making advice (Bhatt & Kankanhalli, 2011).

(1) Through the overall analysis of user reviews, its main content revolves around six elements of milk products: freshness of taste, discounted prices, logistics, customer repurchase, product packaging, nutritional composition;

(2) TF-IDF was able to dig out the words that appeared more frequently in the user reviews. In the word frequency ranking, “delicious” and “date” occupy the top positions in word frequency. In the top 10 rankings, keywords related to taste and texture ranked 1st, 3rd, and 5th, respectively. In contrast, keywords related to freshness ranked 2nd and 4th, respectively, indicating that users focus on the taste and freshness of the milk product. In addition, the 7th and 9th rankings are for “logistics” and “shipping”, indicating that users are particularly concerned about the logistics of the product, in addition to the taste and freshness of the product;

(3) Word2vec can analyze the related words of each keyword to obtain the connection between each word. For example, “date” is related to “freshness”, “shelf life“, “date of production” and so on. “price” is similar to “prices”, “event”, and “economical”;

(4) The knowledge graph drawn by Neo4j can visualize the relevance of each word, visually display the connection between all comments, and categorize the overall content of user reviews into specific categories according to the classification, defining different topics according to the class. For example, words such as “delicious”, “milk quality”, “taste” and their related words point to taste the freshness, “regular clients”, “repurchase”, and “long-term”, and their associated terms point to customer repurchase;

(5) ISM can analyze the interplay between the elements. In tier 3 elements, taste freshness affects discounted prices, customer repurchase, and nutritional composition; in tier 2 elements, product packaging affects discounted prices, and logistics, nutritional composition affects customer repurchase; in tier 1 elements, discounted prices and customer repurchase affect each other and logistics, while price offer involves logistics and logistics express affects customer repurchase. Since milk is a nutritious food that can be consumed directly, the results of this study can be applied to other beverages, fruits and vegetables, and other foods. For manufacturers, the results can be used to improve their products, such as strictly controlling the packaging and paying attention to the degree of freshness of the food, etc. For consumers, it is important to pay attention to the time to consume the products in time to prevent them from exceeding the shelf-life.

Path optimisation

According to the hierarchy diagram in Fig. 4, the elements of layers 3 and 2 points to the producers, while the aspects of layer one end to the e-commerce platform, i.e., the e-commerce platform is the core of the entire hierarchy. Also, it serves as a bridge between the producers and consumers, whose goal is to attract consumers and promote product sales by providing convenient shopping channels and an engaging shopping experience. The smooth operation of an e-commerce platform depends on its operational strategy and the quality of the products supplied by producers (Kolay, 2015).

For producers, ensuring the freshness of their dairy products means adopting appropriate production and storage methods to ensure that products remain fresh and of high-quality during transport and distribution and that hygiene standards and quality requirements are met at all stages. At the same time, producers should ensure that milk products have an excellent nutritional composition and meet the health needs of consumers by adding essential elements and other nutrients to the ingredients following scientific formulation requirements to provide consumers with a balanced choice of products. In addition, manufacturers should also focus on product packaging design. Good packaging not only attracts consumers’ attention and communicates the quality and value of the product to them but also provides the necessary protection to ensure that the product is not damaged during transportation and delivery.

When upstream producers can deliver high-quality products, downstream e-commerce platforms can develop appropriate and sensible marketing strategies to attract purchases and repeat purchases. For example, e-commerce platforms can highlight a product’s freshness and nutritional value and increase its exposure and appeal by showcasing its benefits and detailed descriptions. In addition, e-commerce platforms can promote sustained purchase behavior by actively collecting user feedback and reviews to improve product quality and service and continuously increase user satisfaction.

There is an interdependent relationship between the e-commerce platform and the manufacturer. The manufacturer’s provision of high-quality products is a prerequisite for the smooth operation of the e-commerce platform. In contrast, the effective marketing strategy of the e-commerce platform can facilitate the sale of the manufacturer’s products and user repurchases. Through close cooperation and continuous product quality improvement, both parties have jointly promoted the successful operation of milk products on the e-commerce channel.

Conclusion

In this study, based on the Word2Vec model, the semantic relationship between words is measured in terms of the cosine similarity between word vectors by employing the transformation of words in user comment texts into vectors. Specifically, the word vectors generated by Word2Vec characterize the semantic similarity between words by characterizing them in a multi-dimensional space, and these vectors not only reflect the frequency of occurrence of words, but also reveal their semantic associations in the context. To avoid the interference caused by different word vector lengths, the word vectors will be normalized (i.e., the modulus of the vectors is normalized to unit vectors), to more accurately measure the relative distance between words. This method effectively captures the semantic relationships implied in user comments, providing a more in-depth basis for product optimization and user demand analysis.

However, there are still some limitations in this paper: on the one hand, the research sample only involves user comments on milk products, and user concerns on other categories of products need to be further analyzed; on the other hand, the data source is limited to the Suning eBay Platform, and it does not cover comments on other e-commerce platforms such as Taobao and Jingdong. Therefore, the analysis of user comments on multiple platforms and categories will be a key direction for subsequent research. In addition, future research should further introduce and compare other embedding methods (Pouli et al., 2015), such as transformer-based embedding models (e.g., BERT), to more comprehensively explore the deep semantic relationships in user reviews. This will help assess the applicability and advantages of different embedding methods in user comment analysis and provide a stronger theoretical basis for the selection of text mining techniques.

Supplemental Information

Supplemental Information 1 Data Acquisition

Supplemental Information 2 Computer Code

Supplemental Information 3 Raw Data

We crawled the data of Erie, Mengniu, Junlebao, Ganti Farm, Huang’s Dairy, Shengmu, Anjia, Deya and other brands of milk products in the official website of Suning E-shopping, and crawled a total of 60,367 pieces of data, removing the system’s default comments, blank comments, garbled text and other invalid comments, and finally obtaining the effective The total number of data crawled is 60,367. Before the formal text analysis, it is necessary to use the jieba participle for the participle processing of the comment text. In order to achieve more accurate segmentation results and the accuracy of the subsequent text analysis, the punctuation marks and useless characters (numbers, intonation, etc.) are deactivated, i.e., the results are filtered by words, so as not to affect the extraction of keywords and the analysis of word relevance.

Additional Information and Declarations

Competing Interests

Author Contributions

Data Availability

The authors declare there are no competing interests.

Na Li conceived and designed the experiments, performed the experiments, analyzed the data, performed the computation work, prepared figures and/or tables, organising the team, and approved the final draft.

Yu-Tao Liu performed the experiments, performed the computation work, prepared figures and/or tables, and approved the final draft.

Zhan Chen performed the experiments, prepared figures and/or tables, authored or reviewed drafts of the article, and approved the final draft.

The following information was supplied regarding data availability:

The raw data is available in the Supplemental File.

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
