# Peer review of "Unlocking insights: integrated text mining and interpretive structural modeling for enhanced user review analysis"

_PeerJ Computer Science, doi:10.7717/peerj-cs.2541_

## Round 0.1 · original submission · Major Revisions

Dear authors,

Reviewers have now commented on your paper. They are advising that you revise your manuscript with respect to the basic reporting, experimental design, validity of findings and other comments. If you are prepared to do the necessary revision, I would be happy to reconsider the submission. Furthermore some sentences are too long to read. They should be divided into to or more for readability and comprehensibility (for example the Conclusion section contains only two highly long sentences). Some more recommendations and conclusions should be discussed about the paper considering the experimental results.

Best wishes,

Reviewer 1 ·

Basic reporting

This paper presents a method for extracting effective keywords from massive user reviews of milk products to assist manufacturers and e-commerce platforms in understanding user behavior and preferences, thereby optimizing product design and marketing strategies. The proposed approach combines TF-IDF and Word2Vec text mining techniques to analyze semantic relationships, visualize the relevance of user reviews through knowledge graphs using Neo4j, and explore thematic relationships using the ISM method.

Experimental design

The integration of TF-IDF and Word2Vec techniques to extract and analyze semantic relationships within user reviews is a significant methodological contribution. By combining these two approaches, the paper effectively captures both word frequency and contextual information, offering a comprehensive understanding of consumer sentiment.
The use of Neo4j to create knowledge graphs that visualize the relationships between extracted keywords adds substantial value to the analysis. This approach enables an intuitive understanding of the thematic connections within user reviews, which can be highly beneficial for both manufacturers and e-commerce platforms.
The application of the proposed methodology to real-world data from Suning.com and the categorization of user review elements into hierarchical layers demonstrate the method's practical relevance. This approach allows for actionable insights into consumer behavior, potentially leading to better-targeted marketing strategies and product improvements.
The paper thoroughly examines the relationships and hierarchies between thematic elements using the ISM method.
The use of a web crawler to collect a large dataset of user reviews and the subsequent preprocessing steps, such as word segmentation and filtering, ensure that the analysis is based on high-quality data. This attention to detail enhances the reliability of the study's findings.

Validity of the findings

The paper does not sufficiently compare the proposed method with existing state-of-the-art techniques in the literature, e.g., Personalized multimedia content retrieval through relevance feedback techniques for enhanced user experience 10.1109/ConTEL.2015.7231205. While the novelty of combining TF-IDF and Word2Vec is clear, a more detailed discussion of how this approach outperforms or complements other methods in the field would strengthen the paper.

The scalability of the proposed method, particularly the computational complexity of generating knowledge graphs for large datasets, is not thoroughly addressed. As the size of the dataset increases, the performance and efficiency of the method may be impacted, which could limit its applicability in larger-scale scenarios.
The paper lacks a rigorous quantitative evaluation of the proposed method's performance. Metrics such as accuracy, precision, recall, and F1-score are not provided, making it difficult to assess the effectiveness of the method compared to existing techniques.
How does your approach compare to other state-of-the-art keyword extraction and semantic analysis methods, such as LDA (Latent Dirichlet Allocation) or BERT-based models, in terms of accuracy and computational efficiency?

Additional comments

Can you provide a quantitative evaluation of the proposed method, including metrics such as precision, recall, and F1-score, to better assess its performance?
How does the computational complexity of generating knowledge graphs with Neo4j scale with larger datasets, and what strategies can be employed to mitigate potential performance issues?
In what ways can your method be adapted or extended to analyze user reviews in other product categories or domains beyond milk products?
How does the use of TF-IDF for keyword extraction handle less frequent but potentially significant terms in the reviews, and how might this impact the overall analysis?
Have you considered using alternative text mining techniques, such as topic modeling with LDA, to complement your approach, and if so, how would these techniques compare?
What steps have been taken to ensure that the word segmentation and filtering process does not introduce bias or errors that could affect the accuracy of the subsequent analysis?

Overall this is an interesting research work that needs some revisions in order to be further improved.

Reviewer 2 ·

Basic reporting

The work is interesting since the authors uses NLP embeddings to find relationship between the word in the context of mil e-commerce. However, the work have to be improved to be published:

1) I recommend to show in the results section, how the relation was measured using word2vec? cosine distance? Norm of vectors?
2) The comparison should be performed using other cutting-edges embedding, such as transformers embeddings.
3) Word2Vec and IF-TDF are non-context representation, so, this situation could cause bias in the obtained results.
4) The context should be analyzed to obtain non-biased interpretation of the results.

Experimental design

It is mandatory to use more cutting-edge models to compare the obtained results, since the models used in this work are the beginning of NLP. On the otherside, i recommend to explain the metrics used to compare the results. The dimentional features space should be analyzed, since the context depends on the size of this space.

Finally, i consider that the experiments executed were poor and they have to be riched to reach the publication.

Validity of the findings

Since the methods and experiments should be riched, the conclusion could be biased by the NLP techniques used in this work. It is necessary to show the numeric values and metrics used in the different stages of the research work to validate the obtained results.

---

## Round 0.2 · accepted · Accept

Dear Authors,

Thank you for the revised paper. The reviewers think that you have performed the necessary additions and modifications. Your paper now seems sufficiently improved and acceptable for publication.

Best wishes,

Reviewer 1 ·

Basic reporting

Please refer to the detailed review.

Experimental design

Please refer to the detailed review.

Validity of the findings

Please refer to the detailed review.

Additional comments

The authors have very thoroughly addressed the reviewers' comments providing detailed answers and performing the necessary changes in the manuscript. The quality of the manuscript has been substantially improved and the novelty of the paper has been clearly positioned among the sota.

Reviewer 2 ·

Basic reporting

no comment

Experimental design

The methods need to explained higher. However the work performed is clear.

Validity of the findings

The results are limited, but are coherent with the used methods.

Additional comments

No comment